# Factors associated with preterm birth among mothers who gave birth at public Hospitals in Sidama regional state, Southeast Ethiopia: Unmatched case-control study

**Gossa Fetene** [1]*, **Tamirat Tesfaye**[2], **Yilkal Negesse**[1], **Dubale Dulla**[2]

**1** College of Medicine and Health Sciences, Mizan-Tepi University, Mizan Teferi, Ethiopia, **2** College of Medicine and Health Sciences, Hawassa University, Hawassa, Ethiopia

* Feteneg2119@gmail.com

## Abstract

### Background

Preterm birth remains the commonest cause of neonatal mortality, and morbidity representing one of the principal targets of neonatal health care. Ethiopia is one of the countries which shoulder the highest burden of preterm birth. Therefore, this study was aimed to assess factors associated with preterm birth at public hospitals in Sidama regional state.

### Methods

Facility-based case-control study was conducted at public hospitals in Sidama regional state, from 1st June to 1st September/2020. In this study, a total of 135 cases and 270 controls have participated. To recruit cases and controls consecutive sampling methods and simple random sampling techniques were used respectively. Data were collected using pre-tested structured interviewer-administered questionnaire, and checklist via chart review. Data were entered using EpiData version 3.1 and exported to SPSS version 20 for analysis. Independent variables with P-value < 0.25 in the bivariate logistic regression were candidates for multivariable logistic regression analysis. Finally, statistical significance was declared at P-value < 0.05.

### Results

The response rate was 100%. Rural resident (AOR = 2.034; 95%CI: 1.242, 3.331), no antenatal care service utilization (AOR = 2.516; 95%CI: 1.406, 4.503), pregnancy-induced hypertension (AOR = 2.870; 95%CI: 1.519, 5.424), chronic medical problem during pregnancy (AOR = 2.507; 95%CI: 1.345, 4.676), urinary tract infections (AOR = 3.023; 95%CI: 1.657, 5.513), birth space less than 2 years (AOR = 3.029; 95%CI: 1.484, 6.179), and physical intimate violence (AOR = 2.876; 95%CI: 1.534, 5.393) were significantly associated with preterm birth.

**Data Availability Statement:** All relevant data are within the paper and its Supporting Information files.

**Funding:** The authors received no financial support for the research, authorship, or publication.

**Competing interests:** The authors declared no conflicts of interest concerning the research, authorship, and publication of this article.

**Abbreviations:** ANC, Antenatal Care; FMoH, Federal Minister of Health; HIV, Human Immunodeficiency Virus; LNMP, Last Normal Menstrual Period; PIH, Pregnancy Induced Hypertension; UTIs, Urinary Tract Infections.

## Conclusion

Most of the risk factors of preterm birth were found to be modifiable. Community mobilization on physical violence during pregnancy and antenatal care follow-up are the ground for the prevention of preterm birth because attentive and critical antenatal care screening practice could early identify risk factors. Besides, information communication education about preterm birth prevention was recommended.

## Introduction

Despite tremendous advances in perinatal medicine and the establishment of fetomaternal units,preterm birth remains the leading cause of perinatal mortality and neonatal morbidity representing one of the major targets of obstetrical health care [1]. According to World Health Organization, preterm birth is defined as a delivery that occurs before 37 completed weeks of gestation [2,3]. Out of the 130 million babies born each year globally, approximately 15 million are born prematurely. Of these,60–85% are found concentrated in Africa and South Asia where health systems are weak in access and minimum health services utilization [4]. In Ethiopia, preterm birth ranges from 4.4% to 31.1% [5,6].

Preterm birth is a major healthcare problem causing over 1 million deaths of neonates annually, high rates of morbidity and disability among survivors [1]. Preterm babies predominantly suffer not only from the immediate complication of prematurity, but also long-term complications such as cerebral palsy, intellectual impairment, chronic lung disease, and vision and hearing loss [1,7]. Family, society, and country at large also suffer from the economic burden of preterm birth due to longer hospital stays, neonatal intensive care, and ongoing long-term complex health needs occasioned by the resultant disabilities [7].

Ethiopia is one of the top eight countries that account for the high prevalence of preterm birth in 2014 and the top six countries that contribute nearly two-thirds of all deaths from preterm birth complications worldwide in 2016 [8]. In Ethiopia, 377,000 babies are born too premature each year and 23,100 children under five died due to direct preterm complications [9]. Even if Ethiopia had achieved the millennium development goal (MDG) 4, two years before the targeted year; evidence showed that the pace can't continue as per the plan, and neonatal mortality increased then after [10].

Target 3.2 of the Sustainable development goal (SDG) 3 is to reduce the global neonatal mortality to at least as low as 12 per 1000 live births by 2030 [11]. Also, the Federal Minister of the health of Ethiopian (FMoH), aspires to decrease neonatal mortality to 10 per 1000 live births by 2035 [12]. However, in Ethiopia, the current neonatal mortality is 30 which is too far from the the target set by SDG and FMoH [13]. To achieve SDG and FMoH target better prevention and management of preterm birth and its complications are a key strategy [4]. So, identification of risk factors is crucial to prevent and maintain the management of preterm birth. However, in Ethiopia, few studies were conducted on risk factors associated with preterm birth and showed contradicting findings across different geographical settings and different periods [14–19]. In addition, a study on preterm birth strongly recommends the need for focused, continuous, and comparative studies across different nations and settings. Nevertheless, still, there is a paucity of evidence regarding factors associated with preterm birth at the country level in general and in the study area in particular. As a result, this stated was primarily conducted with the rationale of assessing factors associated with preterm birth in the Sidama

region, Southeast Ethiopia to help policymakers, program designers, and implementers to design appropriate interventions to address the issue of preterm birth.

## Materials and methods

### Study design, setting, and period

An institution-based case-control study was conducted at public hospitals in Sidama regional state from 1[st] June to 1[st] September 2020. Sidama regional state is one of the ten regional states of the country which is located 275 kilometers to the south of Addis Ababa, the capital city of the country. Sidama region has an area of 10,000 km$^2$ of which 97.71% is land, and 2.29% is covered by water. According to the Sidama region health department's 2012 estimation report, it had a total population of 4,369,214 of which 2,201,313 are females and 2,167,901 are males. Around, 1,018,027 females are in reproductive age groups, and the annual estimated prevalence of pregnancies in the region is 5.3%. In the region, there are eighteen hospitals, 132 health centers, and 550 health posts owned by the government. All of the hospitals which provide ANC, labor-delivery, neonatal intensive care unit, and postpartum care as routine service for their catchment population were included in this study.

### Population

All mothers who gave birth at public hospitals in the Sidama region were the source population, and mothers who gave birth at selected public hospitals during the study period were the study populations.

### Case selection criteria

Cases were determined based on gestational age confirmed by the diagnosis of health professionals during admission with the last normal menstrual period or early ultrasound record. Mothers who gave singleton live birth between 28 and 36 completed weeks of gestations and their index neonate were taken to be cases of the study. Mothers who gave singleton live birth between 37 and 41 completed weeks of gestations and their index neonate were taken to be controls of the study.

### Eligibility criteria

Mothers who gave live birth to singleton preterm neonate and their index neonates were **cases** and mothers who gave live birth to singleton term neonate and their index neonates were included as **controls**. Whereas, mothers without a confirmed diagnosis of preterm birth due to disremember their LNMP or who haven't had an early ultrasound record were excluded from the study.

**Sample size determination and sampling technique.** The sample size was calculated using open Epi info version 7 by considering assumption of double population proportions formula with 95% confidence level, 80% power, case to control the ratio of one to two, a minimum detectable odds ratio of 2.15, proportion of controls with exposure of antenatal care less than four (< 4) of 42% [20], 10% non-response rate, and a design effect of 1.5. The required final sample was rounded to 405 with 135 cases and 270 controls.

A multistage sampling technique was employed. One general and three primary hospitals were selected randomly out of three general and fourteen primary hospitals respectively. Also, there was only one comprehensive hospital located in the region and it was included in the study. The total sample size was proportionally distributed to each hospital based on the average number of preterm births recorded in the most recent three months report of each

hospital. Finally, eligible cases were selected consecutively and for each eligible case, controls were selected by using a simple random sampling method until the required sample was achieved (**Fig 1**).

## Operational definitions

**Maternal nutritional assessment** was assessed by measuring the left middle upper arm circumference (MUAC) using non-stretchable MUAC tapes. Most screening programs have used a cut of 21–23 cm (i.e. MUAC <21 cm is severely malnourished, 21-23cm is moderately malnourished, and MUAC >23cm is well malnourished).

**Pregnancy-induced hypertension** was diagnosed as a new onset of hypertension that appears after 20 weeks or more of gestational age of pregnancy with or without proteinuria, which includes gestational hypertension, pre-eclampsia, and Eclampsia.

**Early ultrasound record:** ultrasound result taken until 22 weeks of gestational age [21,22].

**Physical intimate violence**: defined as any act of harm to women physically by the current or former intimate partner or husband [23].

## Data collection tools, procedures, and techniques

The data were collected by face-to-face interviews using a standardized, structured, and pre-tested questionnaire and chart review checklist (S1 Appendix). The questionnaire and checklist were adapted from other similar studies with some contextual modification [23,24]. It

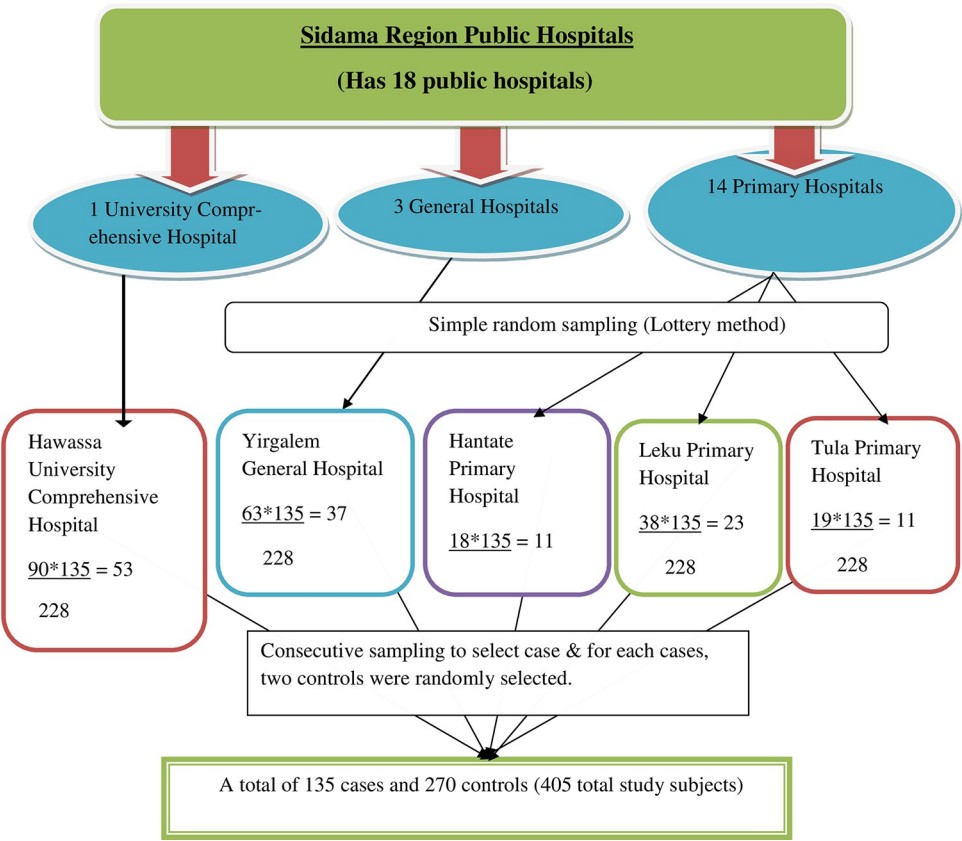

**Fig 1. Schematic presentation of sampling techniques and proportional allocation of cases at public hospitals, Sidama region.**

contains socio-demographic characteristics, obstetric factors, pre-existing medical factors, fetal factors, and physical intimate partner violence. A checklist was used to collect data from the medical record and actual measurements. The checklist was incorporate the gestational age of neonate at birth, sex of neonate, the weight of neonate, height of mother, and current serostatus of the mother.

The gestational age was established based on a certain last menstrual period (LMP) date and/or early pregnancy ultrasound determined date (up to and including 22 completed weeks of gestation). When the LMP and U/S dates had not been correlated, U/S for gestational age assessment was taken in accordance recommendation of the American College of Obstetricians and Gynecologists (ACOG) recommendation [21,22,25]. Those mothers with neither reliable LMP nor early pregnancy U/S date for GA estimation had been excluded. The weight of the neonate was collected by weighted the baby using a calibrated weight scale. A maternal nutritional assessment had been made by using middle upper arm circumference which was measured using an inelastic tape meter. The mothers of both cases and controls were interviewed by the same interviewer in the private room to ensure their privacy and to encourage their communication motives. Data were collected by 10 trained BSc midwives who had been working in the delivery ward. Five BSc midwives were trained and supervised the data collection.

## Data quality assurance

The questionnaire was prepared first in English and translated into the local language Amharic and Sidaamu Afoo and finally, retranslated back to English by a language expert to increase accuracy. Before conducting the study, the questionnaire was pre-tested on 10% of the sample. Based on the pretest, an appropriate modification was made. The one-day training was provided for data collectors and supervisors.

## Data processing and analysis

Collected data were checked for completeness, entered into EpiData version 3.1, coded, and cleaned. After that data were exported to SPSS version 20 for analysis. Descriptive statistics were calculated. Socio-demographic and other profiles of cases and controls were compared using the chi-square test. Statistical significance and strength of association between independent variables and outcome variable were measured using a bivariate logistic regression. Variables with P-value < 0.25 were transferred to multivariable logistic regression to adjust confounders'effects. Correlation between independent variables was assessed to test multicollinearity with a variance inflation factor less than 10 for all variables. Hosmer-Lemeshow test was used to assess model goodness of fit with a P-value > 0.05. The result of the final model was expressed in terms of Adjusted Odd Ratio and 95% confidence intervals and P-value < 0.05 was considered statistically significant.

## Ethical approval and consent to participate

Ethical clearance was obtained from Institutional Review Board (IRB/202/12) at the College of Medicine and Health Sciences of Hawassa University. An official letter of cooperation was obtained from the Department of Midwifery to respective hospital administrators. Informed written permission was obtained from each hospital administrator. After the purpose and objective of the study have been explained, written consent was obtained from each study participant. Participants were informed that participation was voluntary and can withdraw from the study at any time if they were not comfortable with the questionnaire. To keep

confidentiality information was maintained throughout by excluding names or personal identifiers in the questionnaire.

## Results

### Socio-demographic characteristics

A total of 405 study participants had participated. The mean maternal age of the cases and controls were 27.13(±7.19) and 28.75 (±6.51) respectively. The majority of the respondents, 111 (82.2%) of cases and 239(85.9%) of controls were currently married. More than half of both cases 70(51.8%), and controls 195(72.2%) were urban residents (**See Table 1**).

### Obstetrics and social related characteristics

Of the total women who participated in this study, 55.6% of cases and 52.6% of controls have had less than five children. One-fifth of the cases which were 19.3% and 7% of the controls had a birth interval of fewer than two years. More than half of both cases (69.6%) and controls (87%) were utilized ANC at least once in their current pregnancy. Concerning the history of preterm birth, 15.6% of cases and 12.6% of controls had a history of preterm birth. PIH was diagnosed in 28.9% of the cases and 8.5% of controls (**See Table 2**).

### Pre-existing medical problems related characteristics

Of the total study participants, 53(39.3%) of the cases and 102(37.8%) of the control groups were moderately malnourished. Chronic medical problems were diagnosed among 39(28.9%) of the cases and 26(9.6%) of the controls group. UTIs were diagnosed among 44(32.6%) of the cases and 30(11.1%) of the controls group (**See Table 3**).

**Table 1. Socio-demographic characteristics of mothers who gave birth at public hospitals in Sidama region, Southeast Ethiopia, 2020 (n = 405).**

| Variables | Category | Cases (n = 135) | | Controls (n = 270) | | |
|---|---|---|---|---|---|---|
| | | Frequency | % | Frequency | % | P |
| Age | 15–24 | 50 | 37% | 91 | 33.7% | 0.433 |
| | 25–34 | 49 | 36.3% | 90 | 33.3% | |
| | ≥35 | 36 | 26.7% | 89 | 33% | |
| Marital status | Unmarried | 24 | 17.8% | 38 | 14.1% | 0.329 |
| | Married | 111 | 82.2% | 232 | 85.9% | |
| Residence | Urban | 70 | 51.8% | 195 | 72.2% | **0.005** |
| | Rural | 65 | 48.2% | 75 | 27.8% | |
| Educational status | No formal education | 55 | 40.7% | 60 | 22.2% | **0.001** |
| | Primary education | 35 | 25.9% | 109 | 40.4% | |
| | Secondary education | 14 | 10.4% | 54 | 20% | |
| | Diploma and above | 31 | 23% | 47 | 17.4% | |
| Household monthly income | < 1500 | 38 | 28.1% | 55 | 20.4% | 0.138 |
| | 1500–2499 | 46 | 34.1% | 85 | 31.5% | |
| | 2500–3499 | 30 | 22.2% | 86 | 31.9% | |
| | ≥ 3500 | 21 | 15.6% | 44 | 16.3% | |
| Number of family member | < 5 | 18 | 13.3% | 29 | 10.7% | 0.443 |
| | ≥ 5 | 117 | 86.7% | 241 | 89.3% | |

%; Percentage.

**Table 2. Obstetrics and social related characteristics of mothers who gave birth at public hospitals in Sidama region, Southeast Ethiopia, 2020 (n = 405).**

| Variable | Category | Case (n = 135) | | Control (n = 270) | | |
|---|---|---|---|---|---|---|
| | | Frequency | % | Frequency | % | P |
| Gravidity | ≥ 5 | 60 | 44.4% | 128 | 52.6% | 0.573 |
| | < 5 | 75 | 55.6% | 142 | 52.6% | |
| Birth interval of this pregnancy | < 2 year | 26 | 19.3% | 19 | 7% | **0.001** |
| | ≥ 2 year | 109 | 80.7% | 251 | 93% | |
| Antenatal care follow up | No | 41 | 30.4% | 35 | 13% | **0.001** |
| | Yes | 94 | 69.6% | 235 | 87% | |
| First antenatal care started | < 16 weeks | 41 | 43.6% | 119 | 50.6% | 0.250 |
| | ≥ 16 weeks | 53 | 56.4% | 116 | 49.4% | |
| Number of antenatal care | < 4 | 40 | 42.6% | 121 | 51.5% | 0.143 |
| | ≥ 4 | 54 | 57.4% | 114 | 48.5% | |
| Place of ANC follow up | Health post | 18 | 19.1% | 23 | 9.8% | **0.040** |
| | Health center | 41 | 43.6% | 129 | 54.9% | |
| | Gov't hospital, private clinic, and NGO | 35 | 37.2% | 83 | 35.3% | |
| Danger sign of pregnancy advised | No | 8 | 8.5% | 18 | 7.7% | |
| | Yes | 86 | 91.5% | 217 | 92.3% | 0.796 |
| History of preterm birth | Yes | 21 | 15.6% | 34 | 12.6% | 0.412 |
| | No | 114 | 84.4% | 236 | 87.4% | |
| Pregnancy induced Hypertension | Yes | 39 | 28.9% | 23 | 8.5% | **0.001** |
| | No | 96 | 71.1% | 247 | 91.5% | |
| Ante partum bleeding | Yes | 11 | 8.1% | 23 | 8.5% | 0.899 |
| | No | 124 | 91.9% | 247 | 91.5% | |
| Premature rupture of membrane | Yes | 29 | 21.5% | 18 | 6.7% | **0.001** |
| | No | 106 | 78.5% | 252 | 93.3% | |
| Physical intimate violence | Yes | 41 | 30.4% | 23 | 8.5% | **0.001** |
| | No | 94 | 69.6% | 247 | 91.5% | |

**ANC;** Antenatal care follows up, **Gov't hospital**; Government hospital, **NGO**; Nongovernmental organization, %; Percentage.

### Fetal related characteristics

Of the total neonates born, 6.7% of the case group and 5.2% of the control group had birth defects. Concerning to sex of neonates, males accounted for 58.5% of both the case and control group (See **Table 4**).

### Factors associated with preterm birth

On bivariate logistic regression analysis residence, antenatal care utilization, PIH, UTIs, birth spaced less than two years, premature rupture of membrane, a chronic medical problem during pregnancy, and physical intimate violence were associated with preterm birth at P-value < 0.25. Out of these variables; mother's residence, antenatal care utilization, PIH, UTIs, birth spaced less than two years, chronic medical problems during pregnancy, and physical intimate violence was significantly associated with preterm birth on multivariable logistic regression analysis at P-value < 0.05 and 95% confidence level (see **Table 5**).

## Discussion

Even though there have been advancements in perinatal medicine and feto-maternal units, preterm birth remains the leading cause of neonatal mortality & morbidity, take first place for

**Table 3. Medical problems related characteristics of mothers who gave birth at public hospitals in Sidama region, Southeast Ethiopia, 2020 (n = 405).**

| Variables | Category | Case (n = 135) | | Control (n = 270) | | |
|---|---|---|---|---|---|---|
| | | Frequency | % | Frequency | % | P |
| Nutritional status of women in MUAC | SAM | 46 | 34.1% | 86 | 31.9% | 0.736 |
| | Moderate | 53 | 39.3% | 102 | 37.8% | |
| | Normal(>23) | 36 | 26.7% | 82 | 30.4% | |
| Height of women in centimeter | < 150 | 15 | 11.1% | 21 | 7.8% | 0.266 |
| | ≥ 150 | 120 | 88.9% | 249 | 92.2% | |
| Hgb of mother at booking in mg/dl | < 11 | 25 | 18.5% | 39 | 14.1% | 0.289 |
| | ≥ 11 | 110 | 81.5% | 231 | 85.6% | |
| Chronic medical problem | Yes | 39 | 28.9% | 26 | 9.6% | **0.001** |
| | No | 96 | 71.1% | 244 | 90.4% | |
| Urinary tract infections | Yes | 44 | 32.6% | 30 | 11.1% | **0.001** |
| | No | 91 | 67.4% | 240 | 88.9% | |
| Sexually transmitted infections | Yes | 15 | 11.1% | 21 | 7.8% | 0.266 |
| | No | 120 | 88.9% | 249 | 92.2% | |
| HIV status of Mother | Reactive | 14 | 10.4% | 19 | 7% | 0.248 |
| | Non-reactive | 121 | 89.6% | 251 | 93% | |

MUAC; Middle upper arm circumference, SAM; Severe acute malnutrition, %; Percentage.

neonatal intensive care unit admission and longer hospital stay [26,27]. This study aimed to assess factors associated with preterm birth among mothers who gave birth at public hospitals in Sidama regional state, Southeast Ethiopia.

The odds of delivering preterm babies among mothers who lived in rural areas were 2 times more likely than urban residents. This finding is similar to study done in the Amhara region [23], and Axum and Adwa town public hospitals [28]. This might be explained by women who are resided in rural areas are more likely to be exposed to hard physical works like farming which increases the risk of preterm delivery.

The study revealed that mothers who did not utilize antenatal care during their current pregnancy were 2.5 times more likely to deliver preterm babies than mothers who utilized ANC. This result is consistent with a study conducted in Dodola town hospitals, southeast, Ethiopia [15], Kampala, Uganda [29], and a systematic review in East Africa [30]. This might be women who had no ANC follow-up could miss information that is important to prevent, identify, refer, and treat preterm birth promptly in health facilities.

**Table 4. Characteristics of a newborn delivered at public hospitals in Sidama region, Southeast Ethiopia, 2020 (n = 405).**

| Variable | Category | Case (n = 135) | | Control (n = 270) | | |
|---|---|---|---|---|---|---|
| | | Frequency | % | Frequency | % | P |
| Birth defect | Yes | 9 | 6.7% | 14 | 5.2% | 0.544 |
| | No | 126 | 93.3% | 256 | 94.8% | |
| Sex of neonate | Male | 79 | 58.5% | 158 | 58.5% | 1.00 |
| | Female | 56 | 41.5% | 112 | 41.5% | |
| Weight of neonate in gram | < 2500 | 20 | 14.8% | 34 | 12.6% | 0.535 |
| | ≥ 2500 | 115 | 85.2% | 236 | 87.4% | |

%; Percentage.

**Table 5. Bivariate and multivariable logistic regression analysis of factors associated with preterm birth among mothers who gave birth at public hospitals in Sidama regional state, Southeast Ethiopia, 2020.**

| Variable | Category | Case (n = 135) | Control (n = 270) | COR(95%CI) | AOR(95%CI) |
|---|---|---|---|---|---|
| Residence | Rural | 65 | 75 | 2.414(1.571, 3.11) | **2.034(1.242, 3.331)**\* |
|  | Urban | 70 | 195 | 1.00 | 1.00 |
| Antenatal care follow up | No | 41 | 35 | 2.929(1.758, 4.880) | **2.516(1.406, 4.503)**\* |
|  | Yes | 94 | 235 | 1.00 | 1.00 |
| Interval of delivery | <2 year | 26 | 19 | 3.151(1.674,5.933) | **3.029(1.484, 6.179)**\* |
|  | ≥ 2 year | 109 | 251 | 1.00 | 1.00 |
| Pregnancy-induced hypertension | Yes | 39 | 23 | 4.363(2.476,7.689) | **2.870(1.519, 5.424)**\* |
|  | No | 96 | 247 | 1.00 | 1.00 |
| Chronic medical problem in pregnancy | Yes | 39 | 26 | 3.812(2.201,6.605) | **2.507(1.345, 4.676)**\* |
|  | No | 96 | 244 | 1.00 | 1.00 |
| Urinary tract infections | Yes | 44 | 30 | 3.868(2.293,6.526) | **3.023(1.657, 5.513)**\* |
|  | No | 91 | 240 | 1.00 | 1.00 |
| Premature rupture of membrane | Yes | 29 | 18 | 3.830(2.039,7.194) | 1.836(0.867, 3.885) |
|  | No | 106 | 252 | 1.00 | 1.00 |
| Physical intimate violence | Yes | 41 | 23 | 4.684(2.667,8.227) | **2.876(1.534, 5.393)**\* |
|  | No | 94 | 247 | 1.00 | 1.00 |

Key; AOR = Adjusted odd ratio, COR = Crude odd ratio

\* = P-value < 0.05, 1.00 = reference.

The current study depicts that the odds of preterm delivery among mothers who experienced PIH were 2.9 times more likely than mothers who haven't experienced PIH. This finding is in line with a study conducted in Tigray [20], Jimma [24], and Kenyatta national hospital [31]. This might be because hypertension decreases the uteroplacental blood and nutrients transfer which leads to intrauterine growth restriction and/or early placenta dysfunction that cause preterm delivery [32].

Mothers who have experienced UTIs were 3 times more likely to deliver preterm than mothers who haven't experienced UTIs. This result is similar to a study conducted in Nigeria [33] and Kenya [31]. This might be due to UTIs initiate the production of interleukin-1, a known stimulant of labor through the production of prostaglandins from uterine tissue [34].

The odds of preterm delivery among mothers who experienced birth spacing less than two years were 3 times more likely than those who spaced more than two years. This finding is in line with a study conducted in Jimma [24], Northern Ethiopia [35], Axum and Adwa town [28], and the Amhara region [23]. This might be due to mothers having short inter-pregnancy intervals cannot recover from the biological stress imposed by the preceding pregnancy resulting in a reduction of macronutrients supplementation in the maternal body, folate depletion, cervical insufficiency, vertical transmission of infections, incomplete healing of uterine scar and abnormal remodeling of endometrial blood vessels, anemia and maximizing the risk of certain other factors achieving pregnancy outcomes [36,37].

Mothers having a chronic medical problem during their current pregnancy were 2.5 times more likely to deliver preterm babies than mothers who had no medical problems. This finding is consistent with a study conducted in Axum and Adwa town, northern Ethiopia [28] and Yasuj, Iran [38]. This might be due to maternal illnesses reduce the uteroplacental transfer of oxygen and nutrients to the developing fetus which leads to intrauterine growth restriction and/or early placenta dysfunction that cause preterm delivery.

The odds of mothers who experienced physical intimate violence during pregnancy were 2.9 times more likely to deliver preterm babies than mothers who didn't experience physical intimate violence. This finding is in agreement with a study conducted in Iran [39], Vietnam [40], and Tanzania [41]. This might be due to physical violence during pregnancy affect premature delivery because of physical trauma upon the abdomen, uterus, and post-trauma-induced stress which leads to premature onset of labor related to either direct effect or due to corticotrophin-releasing hormone (CRH) [42,43]. This finding has not been supported by a study conducted in Canada [44] and the Amhara region [23]. This difference could be explained by a difference in study design, a discrepancy in the measurement of physical intimate violence, and a difference in socio-demographic characteristics of respondents.

The study revealed that there is no statistically significant association between nutritional status of women during pregnancy and preterm birth. This finding has not been supported by a cross-sectional study conducted in Tigray, Northern Ethiopia [45]. The discrepancy could be explained by a diferrence in the study design, and socio-cultural differences of the study participants.

The study found relevant findings that have paramount importance for preterm birth reduction programs and to plan strategies in the locality. Yet, we would like to assure our reader that few limitations needed to take into account. The study might be prone to recall bias linked to the difficulty of remembering the exact Last normal menstrual period (LNMP) and leads to misclassification bias. However, to solve the problem related to LNMP recall we have used other alternatives like early ultrasound records. The other possible limitation, selection, and information bias might be introduced. To combat this, precise case ascertainment criteria were used in the selection of cases and controls and the same interviewer was used to interview both cases and controls. To attain the accuracy of data the obstetric ultrasound should be done by the same person, and the same machine for all the study units, but it was not possible in our study.

## Conclusion

There were many factors interwoven to affect the occurrence of preterm birth. Preterm birth is more likely to occur in women living in rural areas, with no ANC follow-up, have UTI, PIH, chronic medical problems, low birth spacing, and suffer from physical violence during pregnancy. Community mobilization on physical violence during pregnancy and ANC follow-up are the ground for the prevention of preterm birth because attentive and critical ANC screening practices could early identify the risk factors. Further community-based longitudinal (cohort) studies might explore additional determinants of preterm birth.

## Supporting information

**S1 Appendix. English version questationnaire.**
(DOCX)

**S2 Appendix. Amharic version questionnaire.**
(DOCX)

**S3 Appendix. Sidaamu Afoo version questionnaire.**
(DOCX)

**S1 Data. SPSS statistics data.**
(SAV)

## Acknowledgments

We would like to address our gratitude to our colleagues for the effort they made to enrich our research with important guide and input. We are thankful to our data collectors, supervisors, and study participants. In addition, we would like to acknowledge the staff in all study hospitals for their unlimited cooperation in sharing valuable data when needed which lay a base for the finalization of this research finding.

## Author Contributions

**Conceptualization:** Gossa Fetene, Yilkal Negesse.

**Data curation:** Gossa Fetene, Tamirat Tesfaye, Dubale Dulla.

**Formal analysis:** Gossa Fetene.

**Investigation:** Gossa Fetene.

**Methodology:** Gossa Fetene, Yilkal Negesse, Dubale Dulla.

**Software:** Gossa Fetene, Yilkal Negesse.

**Supervision:** Gossa Fetene, Tamirat Tesfaye, Dubale Dulla.

**Validation:** Gossa Fetene, Tamirat Tesfaye.

**Visualization:** Gossa Fetene, Tamirat Tesfaye.

**Writing – original draft:** Gossa Fetene.

**Writing – review & editing:** Gossa Fetene, Tamirat Tesfaye, Yilkal Negesse, Dubale Dulla.

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
