## [Decision Letter · Decision Letter 0]

14 Aug 2021

PONE-D-21-10270

Factors associated with preterm birth among mothers who gave birth at public Hospitals in Sidama regional state, South East Ethiopia: Unmatched case control study

PLOS ONE

Dear Dr. Fetene,

Thank you for submitting your manuscript to PLOS ONE. After careful consideration, we feel that it has merit but does not fully meet PLOS ONE’s publication criteria as it currently stands. Therefore, we invite you to submit a revised version of the manuscript that addresses the points raised during the review process.

The reviewers have highlighted several aspects of your methods and the contextualisation of your study that require further clarification. Please respond to their comments carefully when preparing your revisions.

We look forward to receiving your revised manuscript.

Kind regards,

Jamie Males

Academic Editor

PLOS ONE

Journal Requirements:

2. Please include additional information regarding the survey or questionnaire used in the study and ensure that you have provided sufficient details that others could replicate the analyses. For instance, if you developed a questionnaire as part of this study and it is not under a copyright more restrictive than CC-BY, please include a copy, in both the original language and English, as Supporting Information.  If the original language is written in non-Latin characters, for example Amharic, Chinese, or Korean, please use a file format that ensures these characters are visible.

6. Please upload a copy of Figure 2, to which you refer in your text on page 5. If the figure is no longer to be included as part of the submission please remove all reference to it within the text.

Reviewers' comments:

Reviewer's Responses to Questions

**Comments to the Author**

1. Is the manuscript technically sound, and do the data support the conclusions?

Reviewer #1: Partly

Reviewer #2: Partly

2. Has the statistical analysis been performed appropriately and rigorously? 

Reviewer #1: Yes

Reviewer #2: I Don't Know

3. Have the authors made all data underlying the findings in their manuscript fully available?

Reviewer #1: Yes

Reviewer #2: Yes

4. Is the manuscript presented in an intelligible fashion and written in standard English?

Reviewer #1: Yes

Reviewer #2: No

5. Review Comments to the Author

Reviewer #1: In this manuscript, the authors investigated factors associated with preterm birth at public hospitals in Sidama regional state, South-east Ethiopia, 2020. They found that risk of PTB is increased in women living in rural areas, with no ANC follow-up, have UTI, PIH, chronic medical problems, low birth spacing, and suffer with physical violence during pregnancy.

The study is well designed, the results are clear, manuscript is well written and clinical significance is high. But following comments/questions should be addressed prior to acceptance.

1. Materials and Methods: The authors described that the definition of PTB is delivery before 37 completed gestational weeks, but the control group of this study consisted of women who delivered at 28-36 weeks of gestation. Is there any reason for excluding extremely early PTB at less than 28 weeks of gestation?

2. Materials and Methods: Calculating gestational age correctly is most important for determining cases and controls (preterm and term birth). In this study, gestational age was confirmed by LMP or early ultrasound record. However, calculating gestational age by LMP may not be correct in many women (especially when their periods were irregular or very long), and recall bias may occur. Since, antenatal check was not done in 30% and 13% of the cases and controls, there may be misclassification bias. This issue should be included in the Discussion section.

4. Discussion: The authors stated that the reason for higher risk of PTB in rural residents may be hard physical works like farming. Is there any evidence for this explanation?

5. The strengths and weaknesses of this study should be discussed in the Discussion section.

Minor issues

1. Introduction, first line: what is ‘tremidinous’? Is it ‘tremendous’?

2. What is MDG 4? The full term of the abbreviations should be presented.

3. Figure 2 is not shown.

Reviewer #2: Review PONE-D-21-10270

Factors Associated with Preterm Birth among Mothers Who Gave Birth at Public Hospitals in Sidama Regional State, South East Ethiopia: Unmatched Case Control Study

This study investigates risk factors of preterm birth in the Sidama region of south east Ethiopia with one of the highest prevalence of preterm births in the world. The aim is to provide insight into risk factors of preterm birth in the region to enable comparison with other nations research on preterm birth.

The authors performed a facility based unmatched case-control study. Preterm birth was defined as singleton live birth between 28 and 36 completed weeks of gestation. Singleton live birth between 37 and 41 completed weeks of gestations were included as controls.

The following aspects were investigated for inclusion as potential risk factor of preterm birth:

Maternal socioeconomic-factors: Maternal age, marital status, residence (Urban or rural), education, household income, number of children. Obstetrics related characteristics: Gravidity, birth interval of this pregnancy, follow up from antenatal care service, first antenatal care started, number of antenatal care, place of ANC follow up, danger sign of pregnancy advised, history of preterm, pregnancy-induced hypertension, ante partum bleeding, premature rupture of membrane. Pre-existing medical problems: MUAC, maternal height (short stature), Hgb of mother at booking in mg/dl, Chronic medical problem, UTI, Sexually transmitted infections, HIV status of Mother. Neonate: Birth defects, sex of neonate, birth weight.

The authors found several factors significantly associated with preterm birth: Rural resident (AOR: 2.034; 95%CI: 1.242, 3.331), no antenatal care service utilization (AOR: 2.516; 95%CI: 1.406, 4.503), pregnancy-induced hypertension (AOR: 2.870; 95%CI: 1.519, 5.424), chronic medical problem during pregnancy (AOR: 2.507; 95%CI: 1.345, 4.676), urinary tract infections (AOR: 3.023; 95%CI: 1.657, 5.513), birth space less than 2 years (AOR: 3.029; 95%CI: 1.484, 6.179), and physical intimate

violence (AOR: 2.876; 95%CI: 1.534, 5.393).

Major

Have cases and controls been selected to obtain equal frequencies of male / female neonate sex in cases and controls? The equal male frequency of 58.5% in cases and controls seems to be somewhat unexpected if controls are drawn by random. Could the authors please see if this is due to method or comment on this?

The discussion could be improved by including something on covariation of risk factors and elaborate on how combinations of these could be less or more important in this region of high prevalence of preterm birth especially compared to in other countries. See several points in discussion below.

Minor

Abstract

In sentence informing about software applied it should be EpiData without hyphen:

“Data were entered using Epi-data version 3.1 and exported to SPSS version 20 for analysis.

Introduction

At page 4 (top section) the authors mentioned some other studies in the field, are any of those possible to include as sources?

“But, in Ethiopia, few studies were conducted on risk factors associated with preterm birth and showed contradicting findings across different geographical settings and different periods.”

Results

Obstetrics related characteristics

Gravidity is categorized into equal to or above 5 and less than 5 which is less informative than primiparous compared to multiparous, is it enough available data to get information on primiparous in the study?

Variable name should be ‘first antenatal care started’ instead of ‘fist antenatal care started”.

Based on frequency of pregnancy-induced hypertension have these pregnancies been assessed for preeclampsia?

Pre-existing medical problems related characteristics

On page 10 the authors use the category ‘chronic medical problems’ without describing in further detail. “Chronic medical problems were diagnosed among 39(28.9%) of the cases and 26(9.6%) of the controls group.”

Is it possible to include some examples of which chronic conditions or medical problems that would classify into this variable?

In table 3 describing maternal pre-existing medical problems, height has an above and below 150 cm threshold, could the authors describe this variable more, is in in relation short stature from to malnutrition?

Fetal related characteristics

See major concern above on equal neonatal male frequency in cases and controls

Factors associated with preterm birth

Table 5, variable pregnancy-induced has a misprint in AOR 2.870(1..519, 5.424)

Discussion

The authors include comparable studies in discussion. Include in the discussion how the results compare to published data also where they deviate. Are there some results that should be expected to associate with preterm birth that show less association than expected in these results?

Please discuss if the results indicate a less prominent role of malnutrition as a risk factor of preterm birth in the region.

For instance, in the data material, the frequency of birth defects seems high. Could many of these be associated to malnutrition (like cleft palate) in a sense that birth defects from malnutrition, even high, is not a prominent risk factors of preterm births in the region? Could this be interpreted in relation to many of the maternal pre-existing medical problems that are malnutrition related and not prominent drivers of preterm birth in the dataset. Could the authors comment on malnutrition and nutritional aspects in relation to preterm birth in the region from the data in the study?

If possible, could the authors elaborate on probable covariation of some of the included risk factors? Is it possible to observe any interaction between rural residency and physical violence or between rural residency and use of antenatal care? Considering rural facilities on sanitation and access to clean water, could possibly rural residency and factors as PPROM and UTI show interaction? Place of ANC follow up seems to indicate some association to rural residence as well.

6. PLOS authors have the option to publish the peer review history of their article (what does this mean?). If published, this will include your full peer review and any attached files.

Reviewer #1: No

Reviewer #2: No

---

## [Author Response · Author response to Decision Letter 0]

27 Aug 2021

Dear Editor and Reviewers,

Thank you very much for your email dated 14th August 2021 incorporating the insight of the editor and reviewer’s comments. On behalf of all authors, I express our gratitude to you for the critical and constructive review that has led to the great improvement of our paper entitled “Factors associated with preterm birth among mothers who gave birth at public Hospitals in Sidama regional state, Southeast Ethiopia: Unmatched case-control study”. We have carefully reviewed the comments given from the editor and all the two reviewers and revised the manuscript accordingly. Our responses are given in a point-by-point manner below for respective editor and reviewers' comments. 

Version 1: PONE-D-21-10270

Date: 8/27/2021

Academic editor comments and a respective author response 

Editor comment 1: Please ensure that your manuscript meets PLOS ONE's style requirements, including those for file naming. The PLOS ONE style templates can be found at https://journals.plos.org/plosone/s/file?id=wjVg/PLOSOne_formatting_sample_main_body.pdf and https://journals.plos.org/plosone/s/file?id=ba62/PLOSOne_ formatting_ sample_ title_authors_affiliations.pdf

Author response: Thanks very much for this comment. The whole parts of the manuscript was updated as per the PLOSE ONE style templates. 

Editor comment 2: Please include additional information regarding the survey or questionnaire used in the study and ensure that you have provided sufficient details that others could replicate the analyses. For instance, if you developed a questionnaire as part of this study and it is not under a copyright more restrictive than CC-BY, please include a copy, in both the original language and English, as Supporting Information. If the original language is written in non-Latin characters, for example Amharic, Chinese, or Korean, please use a file format that ensures these characters are visible.

Author response 2: Thanks very much for this comment. Amharic and Sidaamu Afoo versions of the questionnaires used in this manuscript were attached in the supporting informations (S2 Appendix, and S3 Appendix). 

Editor comment 3: In your Data Availability statement, you have not specified where the minimal data set underlying the results described in your manuscript can be found. PLOS defines a study's minimal data set as the underlying data used to reach the conclusions drawn in the manuscript and any additional data required to replicate the reported study findings in their entirety. All PLOS journals require that the minimal data set be made fully available. For more information about our data policy, please see http://journals.plos.org/plosone/s/data-availability.

Author response 3: Thanks very much for this insightful comment. The underlying data used in the manuscript was attached in the supporting informations (SAV. 1).

Editor comments 4: Your ethics statement should only appear in the Methods section of your manuscript. If your ethics statement is written in any section besides the Methods, please move it to the Methods section and delete it from any other section. Please ensure that your ethics statement is included in your manuscript, as the ethics statement entered into the online submission form will not be published alongside your manuscript.

Author response 4: Thanks very much for this comment. The ethical declaration statement had moved to the method section based on the editor's comment.

Editor comment 5: Please include a separate caption for each figure in your manuscript.

Author response 5: Thanks very much for this comment. The caption had been given for the figure. 

Editor comment 6: Please upload a copy of Figure 2, to which you refer in your text on page 5. If the figure is no longer to be included as part of the submission please remove all references to it within the text.

Author response 6: Thanks a lot for this critical and very insightful comment. Figure 2 had no longer available and it was removed. 

Editor comment 7: Please include captions for your Supporting Information files at the end of your manuscript, and update any in-text citations to match accordingly. Please see our Supporting Information guidelines for more information: http://journals.plos.org/plosone/s/ supporting-information.

Author comment 7: Thanks very much for your insightful comment. The caption had been given for all supporting information files included at the end of the manuscript.

Reviewer #1 comments and an author response

Reviewer #1: In this manuscript, the authors investigated factors associated with preterm birth at public hospitals in Sidama regional state, Southeast Ethiopia, 2020. They found that the risk of PTB is increased in women living in rural areas, with no ANC follow-up, who have UTI, PIH, chronic medical problems, low birth spacing, and suffer with physical violence during pregnancy.

The study is well designed, the results are clear, the manuscript is well written and clinical significance is high. But following comments/questions should be addressed prior to acceptance.

Reviewer comment 1: Materials and Methods: The authors described that the definition of PTB is delivery before 37 completed gestational weeks, but the control group of this study consisted of women who delivered at 28-36 weeks of gestation. Is there any reason for excluding extremely early PTB at less than 28 weeks of gestation?

Author response 1: Thanks very much for this question. Since the study was done in Ethiopia, preterm birth is defined as the delivery of the neonate after 28 weeks of gestation and before 37 completed weeks. If the delivery ended before 28 weeks of gestational age, it was considered as abortion according to the countries guideline (Ethiopia-MOH-Obstetrics-Protocol-2020, page 172). Available at; https://www.scribd.com/document/505962352/Ethiopia-MOH-Obstetrics-Protocol-2020

Reviewer comment 2: Materials and Methods: Calculating gestational age correctly is most important for determining cases and controls (preterm and term birth). In this study, gestational age was confirmed by LMP or early ultrasound records. However, calculating gestational age by LMP may not be correct in many women (especially when their periods were irregular or very long), and recall bias may occur. Since, the antenatal check was not done in 30% and 13% of the cases and controls, there may be misclassification bias. This issue should be included in the Discussion section.

Author response 2: Thanks very much for your insightful comment. It was considered as a limitation of the study which introduced recall bias and explained in the last sentence of the discussion as directed by the reviewer (See Page 19). 

Reviewer comment 3: Discussion: The authors stated that the reason for the higher risk of PTB in rural residents may be hard physical works like farming. Is there any evidence for this explanation?

Author response 3: Thanks a lot for your critical question. In Ethiopia, many women who resided in rural areas are exposed to hard physical works like farming, fetching water, gathering wood. A study conducted by Van Beukering et. al. showed that women who were exposed to hard physical works during pregnancy had higher odds of preterm birth than those who were not exposed. 

Van Beukering, MDM Van Melick, MJGJ Mol, BW Frings-Dresen, et al. Physically demanding work and preterm delivery: a systematic review and meta-analysis. International archives of occupational and environmental health. 2014; 87(8)

Reviewer comment 4: The strengths and weaknesses of this study should be discussed in the Discussion section.

Author response 4: Thanks very much for this comment. The strength and weaknesses of the study had been discussed in the last sentence of the discussion section as commented by the reviewer (See page 19).

Minor comments ;

Reviewer comment #1: Introduction, first line: what is ‘tremidinous’? Is it ‘tremendous’?

Author response 1: Thanks a lot for your critical insight and for correcting our wording error. Yes, it is ‘tremendous’. It was revised and corrected in the sentence. 

Reviewer comment #2: What is MDG 4? The full term of the abbreviations should be presented.

Author response 2: Thanks a lot for your comment. The term MDG 4 is an abbreviation and stands for “Millineum Development Goal 4”. It was revised and stated in the full term (See page 4, line 7 ).

Reviewer comment #3: Figure 2 is not shown. 

Author response 3: Thanks a lot for your insightful comment. Figure 2 had no longer available and it was removed. 

Reviewer #2 comments and an author response: 

Reviewer #2: Review PONE-D-21-10270

Factors Associated with Preterm Birth among Mothers Who Gave Birth at Public Hospitals in Sidama Regional State, South East Ethiopia: Unmatched Case Control Study

This study investigates risk factors of preterm birth in the Sidama region of south east Ethiopia with one of the highest prevalence of preterm births in the world. The aim is to provide insight into risk factors of preterm birth in the region to enable comparison with other nations research on preterm birth. 

Major

Reviewer comment 1: Have cases and controls been selected to obtain equal frequencies of male / female neonate sex in cases and controls? The equal male frequency of 58.5% in cases and controls seems to be somewhat unexpected if controls are drawn by random. Could the authors please see if this is due to method or comment on this?

Author response 1: Thanks very much for your comment. The equal frequencies of male / female neonate sex in cases and controls were due to chance. We used an unmatched case-control study and cases were selected consecutively and for each case, two controls were taken randomly (See page 6). 

The discussion could be improved by including something on covariation of risk factors and elaborate on how combinations of these could be less or more important in this region of high prevalence of preterm birth especially compared to in other countries. See several points in discussion below.

Minor

Reviewer comment 1:

Abstract

In sentence informing about software applied it should be EpiData without hyphen:

“Data were entered using Epi-data version 3.1 and exported to SPSS version 20 for analysis.

Author response 1: Thanks very much for your comment. It was revised and corrected as commented by the reviewer.

Reviewer comment 2:

Introduction

At page 4 (top section) the authors mentioned some other studies in the field, are any of those possible to include as sources?

“But, in Ethiopia, few studies were conducted on risk factors associated with preterm birth and showed contradicting findings across different geographical settings and different periods.”

Author response 2: Thanks a lot for your comment. It was revised and referenced as commented by the reviewer (See page 4, line 15).

Reviewer comment 3:

Results

Obstetrics related characteristics

Gravidity is categorized into equal to or above 5 and less than 5 which is less informative than primiparous compared to multiparous, is it enough available data to get information on primiparous in the study?

Author response 3: Thanks for your insightful comment. Previous studies showed that the risk of preterm birth among grand multiparas was more likely as compared to primipara as well as multipara. Also, we haven’t had enough available data to get information on primiparous in the study to compare the odds of preterm birth among primipara with multipara as well as grand multipara.

Reviewer comment 4: Variable name should be ‘first antenatal care started’ instead of ‘fist antenatal care started”.

Author response 4: Thanks very much for your comment. It was revised and corrected as commented by the reviewer. 

Reviewer comment 5: Based on frequency of pregnancy-induced hypertension have these pregnancies been assessed for preeclampsia?

Author response 5: Thanks a lot for your critical comment. In this research, pregnancy-induced hypertension includes gestational hypertension, pre-eclampsia, and Eclampsia. 

Reviewer comment 6: Pre-existing medical problems related characteristics

On page 10 the authors use the category ‘chronic medical problems’ without describing in further detail. “Chronic medical problems were diagnosed among 39(28.9%) of the cases and 26(9.6%) of the controls group.”

Is it possible to include some examples of which chronic conditions or medical problems that would classify into this variable?

Author response 6: Thanks a lot for this comment. Chronic medical problems were assessed by asking a woman about having been diagnosed with diabetes mellitus, cardiac problem, chronic renal problem, chronic hypertension, chronic liver disease, and others.

Reviewer comment 7: In table 3 describing maternal pre-existing medical problems, height has an above and below 150 cm threshold, could the authors describe this variable more, is in relation short stature from to malnutrition?

Author response 7: Thanks for this critical comment. The classification of the height of the mother was made based on the previous studies. Previous studies showed that the odds of preterm birth among short stature (height less than 150 cm) mothers were more likely as compared to those mothers whose height ≥ 150 cm long. Even though, short stature had a clinical association with malnutrition, in our study it had no statistical association. 

Reviewer comment 8: Fetal related characteristics

See major concern above on equal neonatal male frequency in cases and controls.

Author response 8: Thanks very much for this comment. The equal distribution of male to female neonates was happened due to chance. The selection of cases and controls were discussed under the section sampling technique (See page 6). 

Reviewer comment 9: Factors associated with preterm birth

Table 5, variable pregnancy-induced has a misprint in AOR 2.870(1..519, 5.424)

Author response 9: Thanks very much for your comment. The AOR of the variable pregnancy-induced hypertension was revised and corrected as AOR 2.870(1.519, 5.424).

Reviewer comment 10:

Discussion

The authors include comparable studies in discussion. Include in the discussion how the results compare to published data also where they deviate. Are there some results that should be expected to associate with preterm birth that show less association than expected in these results?

Please discuss if the results indicate a less prominent role of malnutrition as a risk factor of preterm birth in the region.

For instance, in the data material, the frequency of birth defects seems high. Could many of these be associated to malnutrition (like cleft palate) in a sense that birth defects from malnutrition, even high, is not a prominent risk factors of preterm births in the region? Could this be interpreted in relation to many of the maternal pre-existing medical problems that are malnutrition related and not prominent drivers of preterm birth in the dataset. Could the authors comment on malnutrition and nutritional aspects in relation to preterm birth in the region from the data in the study?

Author response 10: Thanks very much for your comment. The variable nutritional status of the mothers had no statistically significant association with preterm birth according to our study, but had a statistically significant association in previously conducted study. And it was revised and disicussed in the discussion section (See page 20, paragraph 1). 

Reviewer comment 11: If possible, could the authors elaborate on probable covariation of some of the included risk factors? Is it possible to observe any interaction between rural residency and physical violence or between rural residency and use of antenatal care? Considering rural facilities on sanitation and access to clean water, could possibly rural residency and factors as PPROM and UTI show interaction? Place of ANC follow up seems to indicate some association to rural residence as well.

Author response 11: Thanks very much for this comment. The correlation between each independent variables was assessed to test multicollinearity with a variance inflation factor (VIF) with a maximum VIF of 1.158 (See page 9, line 15-16).

---

## [Decision Letter · Decision Letter 1]

7 Mar 2022

Factors associated with preterm birth among mothers who gave birth at public Hospitals in Sidama regional state, South East Ethiopia: Unmatched case control study

PONE-D-21-10270R1

Dear Dr. Fetene,

We’re pleased to inform you that your manuscript has been judged scientifically suitable for publication and will be formally accepted for publication once it meets all outstanding technical requirements.

Kind regards,

Ronny Myhre, Ph.D.,

Guest Editor

PLOS ONE

Additional Editor Comments (optional):

Dear Author,

the manuscript has been well worked through and the reviewers comments to the manuscript have been followed up in a very thorough manner.

Reviewers' comments:

Reviewer's Responses to Questions

**Comments to the Author**

1. If the authors have adequately addressed your comments raised in a previous round of review and you feel that this manuscript is now acceptable for publication, you may indicate that here to bypass the “Comments to the Author” section, enter your conflict of interest statement in the “Confidential to Editor” section, and submit your "Accept" recommendation.

Reviewer #1: All comments have been addressed

Reviewer #3: All comments have been addressed

2. Is the manuscript technically sound, and do the data support the conclusions?

Reviewer #1: Yes

Reviewer #3: Yes

3. Has the statistical analysis been performed appropriately and rigorously? 

Reviewer #1: Yes

Reviewer #3: Yes

4. Have the authors made all data underlying the findings in their manuscript fully available?

Reviewer #1: Yes

Reviewer #3: Yes

5. Is the manuscript presented in an intelligible fashion and written in standard English?

Reviewer #1: Yes

Reviewer #3: Yes

6. Review Comments to the Author

Reviewer #1: The authors revised the paper well, and I have no more comment. Although there are are some minor technical errors (e.g., capital/small letters, spaces etc.), the paper now seemed to be suitable for publication.

Reviewer #3: In this manuscript, the authors investigated factors associated with preterm birth at public hospitals in Sidama regional state, Southeast Ethiopia, 2020. The study is well designed, the results are clear, the manuscript is well written and clinical significance is high. After reviewing the other comments from the editors and reviewers, allow me to make a few points.

- Consider including risk factors in keywords

- I understand that ANC (Antenatal care follows up), but I cannot identify that it was named for the first time with its respective acronym before page 12

- In the first paragraph of page 12 the acronym PIH appears, but I cannot identify if it was previously named with its meaning either.

- In the discussion I see that most of the studies that the authors took as a comparison are from the same region or geographic area, as a recommendation they suggest that they take into consideration, for example, developing Latin American countries where they can find very similar results to those studies and would give it a more inclusive character

- In most countries where access to health services is limited and where prenatal care is not started early, determining gestational age can be a real problem. That is why having a first trimester ultrasound is a reasonable alternative for those women who do not remember the date of their last menstrual period, or it is not reliable. While it is true that according to the recommendation of the American College of Obstetricians and Gynecologists in its Committee Opinion No 700 (Committee Opinion No 700: Methods for Estimating the Due Date, Obstetrics & Gynecology: May 2017 - Volume 129 - Issue 5), cited by the authors state that “ A pregnancy without an ultrasound examination that confirms or revises the EDD before 22 0/7 weeks of gestational age should be considered suboptimally dated”. The same committee also establishes that “Ultrasound measurement of the embryo or fetus in the first trimester (up to and including 13 6/7 weeks of gestation) is the most accurate method to establish or confirm gestational age.” Therefore, the authors could consider putting as a limitation that in sonographies after 14 weeks, the margin of error in calculating gestational age could exceed 7 days.

7. PLOS authors have the option to publish the peer review history of their article (what does this mean?). If published, this will include your full peer review and any attached files.

Reviewer #1: No

Reviewer #3: No

---

## [Editor Report · Acceptance letter]

12 Apr 2022

PONE-D-21-10270R1 

Factors associated with preterm birth among mothers who gave birth at public Hospitals in Sidama regional state, Southeast Ethiopia: Unmatched case-control study 

Dear Dr. Fetene:

I'm pleased to inform you that your manuscript has been deemed suitable for publication in PLOS ONE. Congratulations! Your manuscript is now with our production department. 

Kind regards, 

on behalf of

Dr. Ronny Myhre 

Guest Editor

PLOS ONE